# NEIGHBORHOOD-AWARE NEURAL ARCHITECTURE SEARCH

## ABSTRACT

Existing neural architecture search (NAS) methods often return an architecture with good search performance but generalizes poorly to the test setting. To achieve better generalization, we propose a novel neighborhood-aware NAS formulation to identify flat-minima architectures in the search space, with the assumption that flat minima generalize better than sharp minima. The phrase "flat-minima architecture" refers to architectures whose performance is stable under small perturbations in the architecture (*e.g.*, replacing a convolution with a skip connection). Our formulation takes the "flatness" of an architecture into account by aggregating the performance over the neighborhood of this architecture. We demonstrate a principled way to apply our formulation to existing search algorithms, including sampling-based algorithms and gradient-based algorithms. To facilitate the application to gradient-based algorithms, we also propose a differentiable representation for the neighborhood of architectures. Based on our formulation, we propose neighborhood-aware random search (NA-RS) and neighborhood-aware differentiable architecture search (NA-DARTS). Notably, by simply augmenting DARTS (Liu et al., 2019) with our formulation, NA-DARTS finds architectures that perform better or on par with those found by state-of-the-art NAS methods on established benchmarks, including CIFAR-10, CIFAR-100 and ImageNet.

## 1 INTRODUCTION

The process of automatic neural architecture design, also called neural architecture search (NAS), is a promising technology to improve performance and efficiency for deep learning applications (Zoph & Le, 2017; Zoph et al., 2018; Liu et al., 2019). NAS methods typically minimize the validation loss to find the optimal architecture. However, directly optimizing such an objective may cause the search algorithm to overfit to the search setting, *i.e.*, finding a solution architecture with good search performance but generalizes poorly to the test setting. This type of overfitting is a result of the differences between the search and test settings, such as the length of training schedules (Zoph & Le, 2017; Zoph et al., 2018), cross-architecture weight sharing (Liu et al., 2019; Pham et al., 2018), and the usage of proxy datasets during search (Zoph & Le, 2017; Zoph et al., 2018; Liu et al., 2019).

To achieve better generalization, we propose a novel NAS formulation that searches for "flat-minima architectures", which we define as architectures that perform well under small perturbations of the architecture (Figure 1). One example of architectural perturbations is to replace a convolutional operator with a skip connection (identity mapping). Our work takes inspiration from prior work on neural network training, which shows that flat minima of the loss function correspond to network weights with better generalization than sharp ones (Hochreiter & Schmidhuber, 1997). We show that flat minima in the architecture space also generalize better to a new data distribution than sharp minima (Sec. 3.3).

Unlike the standard NAS formulation that directly optimizes single architecture performance, *i.e.*, $\alpha^* = \arg\min_{\alpha \in \mathcal{A}} f(\alpha)$, we optimize the aggregated performance over the neighborhood of an architecture:

$$\alpha^* = \arg\min_{\alpha \in \mathcal{A}} g\left(f(\mathcal{N}(\alpha))\right), \tag{1}$$

where $f(\cdot)$ is a task-specific error metric, $\alpha$ denotes an architecture in the search space $\mathcal{A}$, $\mathcal{N}(\alpha)$ denotes the neighborhood of architecture $\alpha$, and $g(\cdot)$ is an aggregation function (*e.g.*, the mean

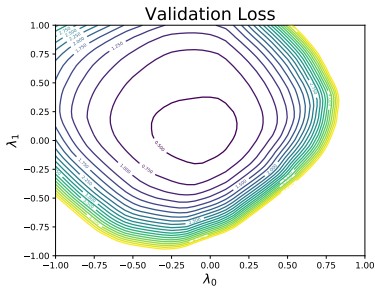

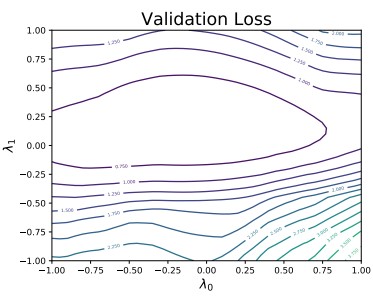

(a) Standard formulation

(b) Neighborhood-aware formulation

Figure 1: Loss landscape visualization of the found architecture. We project architectures (instead of the network weights) onto a 2D plane. The architectures are sampled along two prominent directions (the two axes, $\lambda_0$ and $\lambda_1$), with $(0,0)$ denotes the found architecture. We see that our found architecture (right) is a much flatter minimum than that found with the standard formulation (left). We provide visualization details in the appendix.

function). Note that we overload the notation of the error metric $f(\cdot)$ and define $f(\cdot)$ to return a set of errors when the input is a set of architectures in the neighborhood: $f(\mathcal{N}(\alpha)) = \{f(\alpha') \mid \alpha' \in \mathcal{N}(\alpha)\}$. Common choices for $f(\cdot)$ are validation loss and negative validation accuracy. We will discuss more details of neighborhood $\mathcal{N}(\alpha)$ and aggregation function $g(\cdot)$ in the following text.

To implement our formulation, one must define the neighborhood $\mathcal{N}(\alpha)$ and specify an aggregation function $g(\cdot)$. How to define the neighborhood of an architecture is an open question. One possible method to obtain neighboring architectures is to perturb one or more operations in the architecture and the degree of perturbation defines the scope of the neighborhood. This method can be applied to sampling-based search algorithms, *e.g.*, random search and reinforcement learning. However, it cannot be directly used to generate neighboring architectures for gradient-based search algorithms (*a.k.a*, differentiable NAS), where the neighboring architectures themselves also need to be differentiable with respect to the architecture being learned. To address this issue, we propose a differentiable representation for the neighborhood of architectures, which makes the objective function differentiable and allows us to apply our formulation to gradient-based algorithms, *e.g.*, DARTS (Liu et al., 2019). Properly choosing the aggregation function $g(\cdot)$ can help the search algorithm identify flat minima in the search space. Our choice of $g(\cdot)$ (*e.g.*, mean) is inspired by the definition of the flatness/sharpness of local minima in previous work (Chaudhari et al., 2017; Keskar et al., 2017; Dinh et al., 2017).

We summarize our contributions as follows:

1. We propose a neighborhood-aware NAS formulation based on the flat minima assumption, and demonstrate a principled way to apply our formulation to existing search algorithms, including sampling-based algorithms and gradient-based algorithms. We empirically validate our assumption and show that flat-minima architectures generalize better than sharp ones.

2. We propose a neighborhood-aware random search (NA-RS) algorithm and demonstrate its superiority over the standard random search. On NAS-Bench-201 (Dong & Yang, 2020), NA-RS outperforms the standard random search by $1.48\%$ on CIFAR-100 and $1.58\%$ on ImageNet-16-120.

3. We propose a differentiable neighborhood representation so that we can apply our formulation to gradient-based NAS methods. We augment DARTS (Liu et al., 2019) with our formulation and name the proposed method NA-DARTS. Our NA-DARTS outperforms DARTS by $1.18\%$ on CIFAR-100 and $1.2\%$ on ImageNet, and also performs better than or on par with state-of-the-art NAS methods.

## 2 RELATED WORK

**Flat Minima.** Hochreiter & Schmidhuber (1997) shows that flat minima of the loss function of neural networks generalize better than sharp minima. Flat minima are used to explain the poor generalization of large-batch methods (Keskar et al., 2017; Yao et al., 2018), where large-batch methods are shown to be more likely to converge to sharp minima. Chaudhari et al. (2017) propose an ob-

jective function for training neural networks so that flat minima are preferred during optimization. Their objective can be interpreted as a weighted average of the (transformed) function values of data points around the local minima, which inspires us to consider mean as one of the aggregation functions. Previous work mentioned above focus on flat minima in the network weight space. However, we study flat minima in the architecture space, which is discrete and fundamentally different from the continuous weights studied in previous work. This makes it non-trivial to apply the flat minima idea to NAS.

Zela et al. (2020) observes a strong correlation between the generalization error of the architecture found by DARTS (Liu et al., 2019) and the flatness of the loss function at the found architecture. They propose several regularization strategies to improve DARTS, such as early stopping before the loss curvature becomes too high. Our flat minima assumption is motivated by their observation and our method can be combined with their regularization strategies.

**NAS - Search Algorithm.** Various search algorithms have been applied to solve NAS, including sampling-based and gradient-based algorithms. Representative sampling-based algorithms include random search (Li & Talwalkar, 2019), reinforcement learning (Baker et al., 2017; Zoph & Le, 2017; Zoph et al., 2018; Zhong et al., 2018), Bayesian optimization (Kandasamy et al., 2018; Cao et al., 2019), evolutionary algorithms (Xie & Yuille, 2017; Real et al., 2017; 2019), and sequential model-based optimization (Liu et al., 2018). To make NAS more computationally efficient, weight sharing across architectures is proposed to amortize the training cost of candidate architectures (Pham et al., 2018; Bender et al., 2018). Based on weight sharing, gradient-based algorithms are proposed to directly learn the architecture with gradient descent (Liu et al., 2019; Xie et al., 2019). Our focus is not proposing novel search algorithms but revisiting the standard NAS formulation. Our proposed formulation can be applied to both sampling-based algorithms and gradient-based algorithms.

**NAS - Search Space.** Search space is crucial for the performance of NAS. One of the most widely used search spaces is the cell search space (Zoph et al., 2018), which searches for a cell that can be stacked multiple times to form the entire network. Our proposed neighborhood-aware formulation is agnostic to the search space, and we specifically showcase our formulation on the cell search space.

## 3 NEIGHBORHOOD-AWARE FORMULATION

We propose a neighborhood-aware NAS formulation (Eq. 1) to identify flat minima in the search space. Our formulation builds upon the assumption that flat-minima architectures usually generalize better than sharp ones. In this formulation, the optimal architecture is selected according to the aggregated performance $g\left(f(\mathcal{N}(\alpha))\right)$ of neighbors of an architecture, instead of the standard criterion, *i.e.*, single architecture performance $f(\alpha)$ only. We now introduce the neighborhood definition of an architecture $\mathcal{N}(\alpha)$ and the aggregation function $g(\cdot)$.

### 3.1 NEIGHBORHOOD DEFINITION AND CELL SEARCH SPACE

Formally defining the neighborhood requires a distance metric between architectures, which largely depends on how an architecture is represented and how the search space is constructed. We adopt the cell search space (Zoph et al., 2018) as it has been widely used in recent NAS methods (Liu et al., 2018; 2019). Instead of the entire architecture, we search for a cell that can be stacked multiple times to form the entire architecture. The number of times the cell is stacked and the output layer are manually defined prior to the search.

A cell is defined as a directed acyclic graph (DAG) consisting of $n$ nodes. Each node represents a feature map. Each directed edge $(i, j)(1 \leq i < j \leq n)$ is associated with an operation used to transform the feature map at node $i$, and passes the transformed feature map to node $j$. The feature map at one node is the sum of all the feature maps on the incoming edges to this node: $x^{(j)} = \sum_{(i,j) \in E} \sum_{k=1}^{m} \alpha_k^{(i,j)} o_k(x^{(i)})$, where $E$ denotes the set of edges in the cell, $x^{(i)}$ is the feature map at node $i$, and $o_k$ is the $k^{th}$ operation among the $m$ available operations. $\alpha^{(i,j)}$ is a $m$-dim one-hot vector, indicating the operation choice for edge $(i, j)$. A cell is then represented by a set of variables $\alpha = \{\alpha^{(i,j)}\}$. Note that $\alpha^{(i,j)}$ being a one-hot vector means that only one operation is chosen for edge $(i, j)$. On a side note, the one-hot constraint on $\alpha^{(i,j)}$ can be relaxed in differentiable NAS methods (Liu et al., 2019; Xie et al., 2019).

We define the distance between two cells $\alpha$ and $\alpha'$ as:

$$\text{dist}(\alpha, \alpha') = \sum_{(i,j) \in E} \delta(\alpha^{(i,j)}, \alpha'^{(i,j)}), \tag{2}$$

where $\delta(\cdot, \cdot)$ is the total variation distance between two probability distributions: $\delta(p, q) = \frac{1}{2}||p - q||_1 = \frac{1}{2}\sum_{k=1}^{m} |p_k - q_k|$. Here $p$ and $q$ are both $m$-dim probability distributions. The total variation distance is symmetric and bounded between 0 and 1. It also offers the following property: $\delta(\alpha^{(i,j)}, \alpha'^{(i,j)}) = 0$ implies that the two cells have the same operation at edge $(i, j)$ and $\delta(\alpha^{(i,j)}, \alpha'^{(i,j)}) = 1$ implies that they have different operations at edge $(i, j)$. Note that instead of directly counting the edge differences, we adopt total variation distance to accommodate relaxed $\alpha$ that is later used in differentiable NAS methods (Liu et al., 2019; Xie et al., 2019).

The neighborhood of a cell $\alpha$ is defined as:

$$\mathcal{N}(\alpha) = \{\alpha' \mid \text{dist}(\alpha, \alpha') \leq d\}, \tag{3}$$

where $d$ is a distance threshold. Due to the property of the total variation distance, when $d$ is an integer, the neighborhood contains all the cells that have at most $d$ edges associated with different operations from $\alpha$. For clarification, our definition of neighborhood includes the reference architecture $\alpha$ itself.

## 3.2 AGGREGATION FUNCTION

Given an architecture $\alpha$, the flatness of its neighborhood is determined by how much the performance (*e.g.*, validation loss) of its neighboring architectures varies compared to $\alpha$ itself. Intuitively, when $\alpha$ is a flat minimum, its neighboring architectures should perform similarly to $\alpha$. However, when $\alpha$ is a sharp minimum, the loss of architectures around $\alpha$ increases drastically compared to $\alpha$.

Based on this intuition, we set $g(\cdot)$ as the mean function, since the mean validation loss of architectures around a flat minimum is expected to be lower than those around a sharp minimum. Importantly, minimizing $\text{mean}(f(\mathcal{N}(\alpha)))$ ensures that $\alpha$ is a local minimum and at the same time has a flat neighborhood. For a similar reason, median and max are also valid choices for $g(\cdot)$ to differentiate between flat minima and sharp minima. We provide more discussions of the aggregation function in the appendix.

## 3.3 JUSTIFICATION OF FLAT MINIMA ASSUMPTION

### 3.3.1 FLAT MINIMA GENERALIZE BETTER

Flat minima in the network weight space are shown to generalize better than sharp ones (Hochreiter & Schmidhuber, 1997). However, we focus on flat minima in the architecture space, which is discrete and fundamentally different from the continuous weights studied in previous work. So we conduct experiments on NAS-Bench-201 (Dong & Yang, 2020) to verify that flat minima in the architecture space also generalize better.

NAS-Bench-201 provides a simulated environment for NAS experiments. Using NAS-Bench-201, we search on CIFAR-10 and evaluate the found architectures not only on CIFAR-10, but also on CIFAR-100 and ImageNet-16-120 to better assess the generalization performance of architectures. We select 100 architectures from NAS-Bench-201 that have the lowest validation error on CIFAR-10 to represent local minima in the search space. Next, we show that among these local-minima architectures, flat minima outperform sharp ones, especially on CIFAR-100 and ImageNet-16-120.

We measure the flatness of each local-minimum architecture with its neighborhood variance: the variance of the search-time validation error of its neighboring architectures on CIFAR-10. Based on their neighborhood variance, we divide the 100 architectures into 2 groups: (1) flat minima, which are the 50 architectures with a flat neighborhood (low neighborhood variance), and (2) sharp minima, which are the other 50 architectures with a sharp neighborhood (high neighborhood variance).

We observe that the average search-time validation error of flat minima and sharp minima are almost the same (14.55% and 14.57%). But, as shown in Table 1a, the average test error of flat minima is lower than sharp minima on all three datasets, especially on CIFAR-100 (1.10%) and ImageNet-16-120 (1.24%). This verifies that flat minima generalize better.

Table 1: **Left**: Average test error of flat-minima architectures and sharp-minima architectures. Flat minima consistently outperform sharp minima on all three datasets. **Right**: Kendall's Tau (rank correlation) of the standard criterion $f(\alpha)$ (baseline) and our criterion $g\left(f(\mathcal{N}(\alpha))\right)$. Our criterion gives a more accurate ranking of architectures on all three datasets.

| (a) | | | | (b) | | | |
| --- | --- | --- | --- | --- | --- | --- | --- |
| | CIFAR-10 | CIFAR-100 | ImageNet-16-120 | | CIFAR-10 | CIFAR-100 | ImageNet-16-120 |
| Flat minima | **6.23** | **28.90** | **55.17** | Baseline | $0.66 \pm 0.03$ | $0.66 \pm 0.02$ | $0.64 \pm 0.03$ |
| Sharp minima | 6.66 | 30.00 | 56.41 | Ours | $\mathbf{0.76 \pm 0.03}$ | $\mathbf{0.77 \pm 0.03}$ | $\mathbf{0.74 \pm 0.03}$ |

### 3.3.2 Aggregated Performance Gives a Better Ranking of Architectures

Based on the flat minima assumption, our neighborhood-aware formulation suggests using the aggregated performance $g\left(f(\mathcal{N}(\alpha))\right)$ as the criterion to select optimal architectures, instead of the standard criterion $f(\alpha)$. The selection criterion determines whether we can obtain an accurate ranking of candidate architectures during search, and further determines the performance of found architectures. We show that our criterion $g\left(f(\mathcal{N}(\alpha))\right)$ ranks architectures more accurately than $f(\alpha)$.

We evaluate the ranking estimated by our criterion $g\left(f(\mathcal{N}(\alpha))\right)$ or the standard criterion $f(\alpha)$ on NAS-Bench-201. Following Yu et al. (2020), we use the the Kendall's Tau metric (the higher the better) to measure the correlation between the estimated ranking and ground truth ranking of architectures. As the ground truth ranking is specific to each dataset, we evaluate the estimated ranking on the three datasets separately. From Table 1b, we see that our criterion $g\left(f(\mathcal{N}(\alpha))\right)$ ($g(\cdot) = \text{mean}$) ranks architectures much more accurately than the standard criterion $f(\alpha)$. Due to space constraint, we include experimental details and more results of other aggregation functions in the appendix.

## 4 Neighborhood-Aware Search Algorithms

We propose neighborhood-aware random search and neighborhood-aware DARTS by applying our formulation to random search (sampling-based) and DARTS (gradient-based), respectively.

### 4.1 Neighborhood-Aware Random Search

When applying our formulation to random search, we only need to change the criterion of selecting optimal architectures from $f(\alpha)$ to the aggregated performance $g\left(f(\mathcal{N}(\alpha))\right)$. At each step, we randomly sample an architecture $\alpha$ and compute its aggregated performance $g\left(f(\mathcal{N}(\alpha))\right)$, and choose the one with the best aggregated performance as our solution. We provide a detailed algorithm sketch of neighborhood-aware random search (NA-RS) in the appendix.

In practice, the entire neighborhood may be large. Instead of using all the neighbors, we sample a subset of $n_{\text{nbr}}$ neighboring architectures from the neighborhood. In our implementation, we always include the reference architecture itself in the sampled subset.

Note that since NA-RS evaluates a neighborhood of architectures at each step, for fair comparison, we allow the standard random search (baseline) to run for more steps such that the two methods evaluate the same number of architectures during search. Specifically, if our NA-RS searches for $T$ steps, the standard random searches for $T \cdot n_{\text{nbr}}$ steps.

While we only present the application of our formulation to random search, the formulation is also applicable to other sampling-based search algorithms, such as reinforcement learning (RL) and Bayesian optimization (BO). Similar to NA-RS, when applying our formulation to RL or BO, we only need to define the reward signal in RL or the objective function in BO as the aggregated performance $g\left(f(\mathcal{N}(\alpha))\right)$. Other components in RL or BO remain unchanged.

### 4.2 Neighborhood-Aware Differentiable Search

We now present how to apply our formulation to differentiable NAS methods. The key in these methods (Liu et al., 2019; Xie et al., 2019; Chen et al., 2019) is to make the objective $f(\alpha)$ differentiable with respect to the architecture $\alpha$ such that one can optimize $\alpha$ with gradient descent.

Similar to the case of random search, our formulation changes the objective from $f(\alpha)$ to $g\left(f(\mathcal{N}(\alpha))\right)$. With this change, the differentiability of $g\left(f(\mathcal{N}(\alpha))\right)$ is not guaranteed. Therefore, we propose a differentiable neighborhood representation for $\mathcal{N}(\alpha)$ and set the aggregation

---

**Algorithm 1** Neighborhood-Aware DARTS

---

**Input**: Number of steps $T$. Number of neighbors $n_{\text{nbr}}$. Initial architecture $\alpha$ and weights $w$.
**for** $t = 1, 2, \ldots, T$ **do**
    Sample a batch of training data $X_{\text{train}}$ and a batch of validation data $X_{\text{val}}$.
    Sample $n_{\text{nbr}}$ neighboring architectures of $\alpha$: $\mathcal{N}(\alpha)$.
    Compute $\nabla_\alpha \frac{\sum_{\alpha' \in \mathcal{N}(\alpha)} \mathcal{L}_{\text{val}}(w, \alpha')}{|\mathcal{N}(\alpha)|}$ on $X_{\text{val}}$; update $\alpha$ by descending $\nabla_\alpha \frac{\sum_{\alpha' \in \mathcal{N}(\alpha)} \mathcal{L}_{\text{val}}(w, \alpha')}{|\mathcal{N}(\alpha)|}$.
    Compute $\nabla_w \mathcal{L}_{\text{train}}(w, \alpha)$ on $X_{\text{train}}$; update $w$ by descending $\nabla_w \mathcal{L}_{\text{train}}(w, \alpha)$.
**end for**
Derive the final architecture based on the learned $\alpha$.

---

function $g(\cdot)$ to be $\mathrm{mean}$ ($g$ can also be other differentiable functions). This makes $g\left(f(\mathcal{N}(\alpha))\right)$ differentiable and allows us to simply adopt prior gradient estimation techniques, *e.g*, the continuous relaxation in DARTS (Liu et al., 2019) or Gumbel-Softmax in SNAS (Xie et al., 2019), to derive the gradient of $g\left(f(\mathcal{N}(\alpha))\right)$. Other parts in the original diffferentiable NAS methods remain the same.

Specifically, we augment DARTS (Liu et al., 2019) with our formulation and adopt the continuous relaxation in DARTS to estimate the gradient. Therefore, we name our method neighborhood-aware DARTS (NA-DARTS). Note that our formulation is also applicable to other differentiable NAS methods, such as SNAS (Xie et al., 2019) and P-DARTS (Chen et al., 2019).

### 4.2.1 NEIGHBORHOOD-AWARE DARTS

We first briefly review DARTS and then introduce the formulation of our NA-DARTS.

**DARTS.** DARTS relaxes the discrete search space to be continuous so that the gradient of the validation loss with respect to the architecture $\alpha$ can be estimated, allowing optimizing $\alpha$ with gradient descent. Concretely, $\alpha^{(i,j)}$ is relaxed from a discrete one-hot vector to a continuous distribution, and is parameterized as the output of the softmax function: $\alpha_k^{(i,j)} = \frac{\exp(\beta_k^{(i,j)})}{\sum_{k=1}^m \exp(\beta_k^{(i,j)})}$, where $m$ is the number of available operations and $\beta = \{\beta_k^{(i,j)}\}$ is the set of continuous logits to be learned. DARTS formulates NAS as the following bilevel optimization problem:

$$\min_\alpha \mathcal{L}_{\text{val}}(w^*(\alpha), \alpha) \qquad \text{s.t.} \ w^*(\alpha) = \arg\min_w \mathcal{L}_{\text{train}}(w, \alpha), \tag{4}$$

where $w$ denotes network weights, $w^*(\alpha)$ denotes the weights minimizing the training loss of architecture $\alpha$. $\mathcal{L}_{\text{train}}(w, \alpha)$ and $\mathcal{L}_{\text{val}}(w, \alpha)$ are the training loss and validation loss of architecture $\alpha$ with weights $w$, respectively.

**NA-DARTS.** We augment DARTS with our neighborhood-aware formulation:

$$\min_\alpha g(\{\mathcal{L}_{\text{val}}(w^*(\alpha'), \alpha') \mid \alpha' \in \mathcal{N}(\alpha)\}) \qquad \text{s.t.} \ w^*(\alpha') = \arg\min_w \mathcal{L}_{\text{train}}(w, \alpha'), \tag{5}$$

where $\mathcal{N}(\alpha)$ is the neighborhood of architecture $\alpha$ and $g(\cdot)$ is an aggregation function.

An outline of the proposed NA-DARTS algorithm can be found in Algorithm 1. We first describe how to represent the neighboring architecture $\alpha'$ as a differentiable function of $\alpha$ and, then discuss the gradient estimation for specific choices of $g(\cdot)$.

### 4.2.2 DIFFERENTIABLE NEIGHBORHOOD REPRESENTATION

When the one-hot constraint on $\alpha$ is relaxed, the neighborhood contains an infinite number of neighboring architectures. We propose a method to sample a finite number of architectures from the neighborhood. Importantly, our method allows each sampled neighbor $\alpha'$ to be differentiable with respect to the reference architecture $\alpha$.

We generate neighboring architectures of $\alpha$ by perturbing the operations associated with the edges in $\alpha$. We randomly sample $d$ edges to be perturbed from the edge set $E$ of $\alpha$ and leave the operation choice for the remaining edges unchanged. This implies that the distance between $\alpha$ and the neighboring architecture $\alpha'$ is at most $d$, thus as defined in Eq. 3, $\alpha'$ falls into the neighborhood of $\alpha$. Next, we present how to represent $\alpha'$ as a differentiable function of $\alpha$.

Let edge $(i, j)$ be an edge to be perturbed. Let $q^{(i,j)}$ be a $m$-dim real-valued noise vector satisfying the following condition: $|q_k^{(i,j)}| \leq \epsilon(0 < \epsilon < 1)$ and $\alpha_k^{(i,j)} + q_k^{(i,j)} \geq 0$ for all $k(1 \leq k \leq m)$. $\epsilon$ is the threshold of the noise. We randomly sample a noise vector $q^{(i,j)}$ and $\alpha'^{(i,j)}$ is computed as:

$$\alpha_k'^{(i,j)} = \frac{\alpha_k^{(i,j)} + q_k^{(i,j)}}{\sum_{k=1}^{n}(\alpha_k^{(i,j)} + q_k^{(i,j)})}. \tag{6}$$

Repeating the process for each edge to be perturbed will result in a neighboring architecture $\alpha'$, which is differentiable with respect to $\alpha$. Different noise vectors are sampled for different edges. We term the representation of $\alpha'$ in Eq. 6 as the *additive representation* of neighboring architectures.

With the additive representation, we can sample a set of neighboring architectures of $\alpha$ and the sampled architectures are differentiable with respect to $\alpha$. In practice, we uniformly sample $n_{\text{nbr}}$ neighboring architectures from the neighborhood and always include $\alpha$ itself in the sampled set.

### 4.2.3 GRADIENT ESTIMATION

After sampling a finite set of neighboring architectures, we compute the validation loss of each individual architecture $\alpha'$, where we use the current network weights $w$ as an approximation of $w^*(\alpha')$. Then we pass the set of the validation losses to the aggregation function $g(\cdot)$.

As discussed before, the aggregation function $g(\cdot)$ needs to be differentiable, which immediately rules out median. We choose mean over max due to its superior empirical performance. We note that when using max, Eq. 5 becomes a minimax optimization problem and one can approximate the gradient of the objective using Danskin's Theorem (Danskin, 1967). For completeness, we provide details of using max in the appendix.

## 5 EXPERIMENTAL RESULTS

### 5.1 NEIGHBORHOOD-AWARE RANDOM SEARCH

We validate our NA-RS on NAS-Bench-201 (Dong & Yang, 2020). Same as the experimental setup in Sec. 3.3, we search on CIFAR-10 and evaluate on CIFAR-10 (Krizhevsky et al., 2009), CIFAR-100 (Krizhevsky et al., 2009), and ImageNet-16-120 (Dong & Yang, 2020). The number of search steps $T$ in NA-RS is set to 100. We set the distance threshold $d$ to 1 and sample 10 neighbors ($n_{\text{nbr}} = 10$) at each step. Please see the appendix for more details.

As shown in Table 2, NA-RS consistently outperform RS on all three datasets, which validates our neighborhood-aware formulation. Notably, NA-RS outperforms RS by $1.48\%$ on CIFAR-100 and $1.58\%$ ImageNet-16-120. Note that the cell search space typically has a narrow performance range (Yang et al., 2020), so the improvement brought by our NA-RS is non-trivial. We include the ablation study of $n_{\text{nbr}}$ and the aggregation function in NA-RS in the appendix.

### 5.2 NEIGHBORHOOD-AWARE DARTS

Following the experimental setup in DARTS (Liu et al., 2019), we search on CIFAR-10 (Krizhevsky et al., 2009) and evaluate on three datasets: CIFAR-10 (Krizhevsky et al., 2009), CIFAR-100 (Krizhevsky et al., 2009) and ImageNet (Russakovsky et al., 2015). The performance on

Table 2: Test error of NA-RS and the standard random search (RS). NA-RS consistently outperforms RS on all three datasets under the same computational budget.

|  | CIFAR-10 | CIFAR-100 | ImageNet-16-120 |
|---|---|---|---|
| Random Search (RS) | $6.39 \pm 0.32$ | $29.81 \pm 0.44$ | $56.30 \pm 1.08$ |
| NA-RS (Ours) | $\mathbf{6.20 \pm 0.35}$ | $\mathbf{28.33 \pm 1.22}$ | $\mathbf{54.72 \pm 0.96}$ |

Table 3: Test error of NA-DARTS and DARTS on CIFAR-10, CIFAR-100 and ImageNet. Our NA-DARTS consistently outperforms DARTS on all three datasets.

| Method | Top-1 Test Error (%) | | | Params (M) | |
|---|---|---|---|---|---|
| | CIFAR-10 | CIFAR-100 | ImageNet | CIFAR | ImageNet |
| DARTS 1st (Liu et al., 2019) | $2.90 \pm 0.25$ | $17.66 \pm 0.83$ | - | 2.9 | - |
| DARTS 2nd (Liu et al., 2019) | $2.70 \pm 0.08$ | $17.72 \pm 0.61$ | 26.7 | 2.9 | 4.7 |
| NA-DARTS (Ours) | $\mathbf{2.63 \pm 0.12}$ | $\mathbf{16.48 \pm 0.13}$ | $\mathbf{25.5}$ | 3.2 | 4.8 |

Table 4: Comparison with state-of-the-art NAS methods on CIFAR-10 and CIFAR-100. Our NA-DARTS achieves the lowest test error on CIFAR-100. As all the architectures are searched on CIFAR-10, this shows that architectures found by NA-DARTS generalize better.

| Method | Test Error (%) | | Params (M) | Search Cost (GPU days) | Search Method |
|---|---|---|---|---|---|
| | CIFAR-10 | CIFAR-100 | | | |
| NASNet-A (Zoph et al., 2018) | 2.65 | 17.10[*] | 3.3 | 1800 | RL |
| AmoebaNet-A (Real et al., 2019) | 2.84[*] | 17.16[*] | 3.2 | 3150 | Evolution |
| PNAS (Liu et al., 2018) | 2.95[*] | 17.29[*] | 3.2 | 225 | SMBO |
| ENAS (Pham et al., 2018) | 2.54[*] | 17.18[*] | 3.9 | 0.5 | RL |
| SNAS (Xie et al., 2019) | $2.85 \pm 0.02$ | 18.25[*] | 2.8 | 1.5 | Gradient |
| P-DARTS (Chen et al., 2019) | 2.50 | 16.55 | 3.4 | 0.3 | Gradient |
| PC-DARTS (Xu et al., 2020) | $2.57 \pm 0.07$ | 16.74[*] | 3.6 | 0.1 | Gradient |
| DARTS+ (Liang et al., 2019) | 2.72[*] | 16.85[*] | 4.3 | 0.6 | Gradient |
| DARTS 1st (Liu et al., 2019) | $2.90 \pm 0.25$ | $17.66 \pm 0.83$ | 2.9 | 0.3 | Gradient |
| DARTS 2nd (Liu et al., 2019) | $2.70 \pm 0.08$ | $17.72 \pm 0.61$ | 2.9 | 1.0 | Gradient |
| NA-DARTS (Ours) | $2.63 \pm 0.12$ | $\mathbf{16.48 \pm 0.13}$ | 3.2 | 1.1 | Gradient |

[*] We train the reported architecture following the training setup in DARTS (Liu et al., 2019).

Table 5: Comparison with state-of-the-art NAS methods on ImageNet. Our NA-DARTS obtains the second lowest test error on ImageNet. We expect further improvement since our contribution is orthogonal to other extensions of DARTS (*e.g.*, P-DARTS, PC-DARTS and DARTS+).

| Method | Test Error (%) | | Params (M) | +× (M) | Method | Test Error (%) | | Params (M) | +× (M) |
|---|---|---|---|---|---|---|---|---|---|
| | Top-1 | Top-5 | | | | Top-1 | Top-5 | | |
| DARTS (Liu et al., 2019) | 26.7 | 8.7 | 4.7 | 574 | AmoebaNet-A (Real et al., 2019)[*] | 27.0 | 8.9 | 5.0 | 584 |
| P-DARTS (Chen et al., 2019)[*] | 25.3 | 8.1 | 4.9 | 557 | NASNet-A (Zoph et al., 2018) | 26.0 | 8.4 | 5.3 | 564 |
| PC-DARTS (Xu et al., 2020)[*] | 25.7 | 8.3 | 5.3 | 586 | ENAS (Pham et al., 2018)[*] | 26.1 | 8.6 | 5.2 | 576 |
| DARTS+ (Liang et al., 2019)[*] | 26.4 | 8.5 | 5.0 | 586 | PNAS (Liu et al., 2018) | 25.8 | 8.1 | 5.1 | 588 |
| NA-DARTS (Ours) | 25.5 | 8.2 | 4.8 | 557 | SNAS (Xie et al., 2019) | 27.3 | 9.2 | 4.3 | 522 |

[*] We train the reported architecture following the training setup in DARTS (Liu et al., 2019).

CIFAR-100 and ImageNet are more important, which reflects how well the found architecture can generalize to new datasets. For our NA-DARTS, we sample 10 neighboring architectures at each step, *i.e.*, $n_{\mathrm{nbr}} = 10$. Complete experimental details and ablation results are included in the appendix.

We first compare our NA-DARTS with DARTS. This comparison directly verifies the effectiveness of our neighborhood-aware formulation. As shown in Table 3, NA-DARTS consistently outperforms DARTS on all three datasets. Notably, NA-DARTS outperforms DARTS by $1.18\%$ on CIFAR-100 and $1.2\%$ on ImageNet. Note that the cell search space used in DARTS has a narrow performance range (Yang et al., 2020). For example, the top-1 error on CIFAR-100 mostly fall around $17\%$. So the performance gap between our NA-DARTS and DARTS is non-trivial. We also compare NA-DARTS and DARTS using a different search space in the appendix and observe a bigger gap.

NA-DARTS also outperforms or performs on par with other state-of-the-art NAS methods (Table 4 & 5). NA-DARTS obtains the lowest test error on CIFAR-100 and the second lowest on ImageNet among state-of-the-art NAS methods. Note that P-DARTS, PC-DARTS and DARTS+ are all extensions of DARTS and the neighborhood-aware formulation is also applicable to them. Their ideas to improve DARTS, *e.g.*, gradually increasing search depth in P-DARTS and the partial-channel connection idea in PC-DARTS, can all be combined with our method for better performance. Therefore, our improvement is complementary to theirs in reference to DARTS. We provide further results of applying our formulation to PC-DARTS in the appendix.

## 6   CONCLUSIONS

To achieve better generalization, we propose a novel neighborhood-aware NAS formulation, based on the assumption that flat-minima architectures generalize better than sharp ones. Our formulation provides a new perspective for NAS that one should use the aggregated performance over the neighboorhood as the criterion to select optimal architectures. We also demonstrate a principled way to apply our formulation to existing search algorithms and propose two practical search algorithms NA-RS and NA-DARTS. Extensive experiments on CIFAR-10, CIFAR-100 and ImageNet validate the flat minima assumption, and demonstrate the significance of our formulation and algorithms.

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

## A  GENERALIZATION TEST ON A NEW SEARCH SPACE

Zela et al. (2020) finds that on a wide range of search spaces, while DARTS (Liu et al., 2019) can successfully minimizes the validation loss during search, the found architectures are usually degenerated and generalize poorly to the test setting. To further validate our NA-DARTS, we conduct experiments on the search space suggested by Zela et al. (2020) and show that in this new search space, NA-DARTS can still find architectures that generalize much better than DARTS.

The new search space is a subset of the original DARTS search space. The new search space is exactly the same as the original search space, except that it only considers three candidate operations, including $3 \times 3$ separable convolution, skip connection, and the zero operation. Following Zela et al. (2020), we refer to the new search space as 'S3 search space'.

We search architectures from the S3 search space on CIFAR-10 using DARTS (Liu et al., 2019), DARTS-ES (Zela et al., 2020) or our NA-DARTS, and then evaluate the found architecture on both CIFAR-10 and CIFAR-100. DARTS-ES is DARTS with an early stopping criterion based

Table 6: Test error of architectures found from the S3 search space on CIFAR-10 and CIFAR-100. **Top**: our NA-DARTS significantly outperforms DARTS on both CIFAR-10 and CIFAR-100. **Bottom**: applying our formulation to PC-DARTS yields further improvement. Our NA-PC-DARTS outperforms PC-DARTS by $0.72\%$ on CIFAR-100.

|                            | CIFAR-10           | CIFAR-100          |
| -------------------------- | ------------------ | ------------------ |
| DARTS (Liu et al., 2019)   | $4.13 \pm 0.98$    | $22.49 \pm 2.62$   |
| DARTS-ES (Zela et al., 2020) | $3.71 \pm 1.14$  | $-$                |
| NA-DARTS (Ours)            | $\mathbf{2.97 \pm 0.18}$ | $\mathbf{18.86 \pm 0.49}$ |
| PC-DARTS (Xu et al., 2020) | $\mathbf{2.66 \pm 0.14}$ | $17.38 \pm 0.45$ |
| NA-PC-DARTS (Ours)         | $2.69 \pm 0.08$    | $\mathbf{16.66 \pm 0.39}$ |

on the dominant eigenvalue of the Hessian of the validation loss. We summarize the performance of DARTS, DARTS-ES and NA-DARTS in the top half in Table 6. We see that our NA-DARTS outperforms both DARTS and DARTS-ES on CIFAR-10. Notably, NA-DARTS outperforms DARTS by $3.63\%$ on CIFAR-100, which demonstrates that architectures found by NA-DARTS generalize much better than those found by DARTS. The performance of DARTS-ES on CIFAR-10 is from Zela et al. (2020). We do not report the test error of DARTS-ES on CIFAR-100 because architectures found by DARTS-ES on CIFAR-10 are unavailable to us.

In the main text, we mention that our contribution is orthogonal to other extensions of DARTS (Chen et al., 2019; Xu et al., 2020) and we expect better performance after applying our formulation to them. We propose NA-PC-DARTS by applying our neighborhood-aware formulation to PC-DARTS (Xu et al., 2020), and obtain further improvement. We evaluate PC-DARTS and our NA-PC-DARTS on the S3 search space and report the results in the bottom half in Table 6.

We notice that NA-PC-DARTS performs similarly to PC-DARTS on CIFAR-10. We would like to emphasize that the DARTS search space (a superset of the S3 search space) has a narrow performance range as verified in Yang et al. (2020) and the test error of most NAS methods on CIFAR-10 are within $[2.5\%, 3.0\%]$. So we focus more on the performance on CIFAR-100. Our NA-PC-DARTS outperforms PC-DARTS by $0.72\%$ on CIFAR-100. Note that all the architectures are searched on CIFAR-10, this shows that architectures found by our NA-PC-DARTS generalize better than those found by PC-DARTS. We obtain further improvement after applying our formulation to PC-DARTS.

## B    AGGREGATION FUNCTION

### B.1    MORE CHOICES FOR AGGREGATION FUNCTION

Our formulation aims to identify flat minima in the search space based on the aggregated performance $g\left(f(\mathcal{N}(\alpha))\right)$ over the neighborhood. The aggregation function $g(\cdot)$ needs to be properly set such that minimzing $g\left(f(\mathcal{N}(\alpha))\right)$ results in an architecture $\alpha$ that is a local minimum and at the same time has a flat neighborhood.

The flatness of the neighborhood of $\alpha$ is determined by how much the performance (*e.g.*, validation loss) of its neighboring architectures varies compared to $\alpha$ itself. Intuitively, when $\alpha$ is a flat minimum, its neighboring architectures should perform similarly to $\alpha$. However, when $\alpha$ is a sharp minimum, the loss of architectures around $\alpha$ increases drastically compared to $\alpha$. Although the formal definition of flatness or sharpness of a local minimum is not exactly the same in previous work (Hochreiter & Schmidhuber, 1997; Chaudhari et al., 2017; Keskar et al., 2017; Dinh et al., 2017; Yao et al., 2018), they all share this intuition.

We discuss possible choices for the aggregation function $g(\cdot)$:

- mean, median or max.

  The architectures around a sharp minimum tend to high much higher loss compared to this minimum. Therefore, the mean validation loss of architectures around a flat minimum is expected to be lower than those around a sharp minimum. Minimizing $\mathrm{mean}\left(f(\mathcal{N}(\alpha))\right)$ encourages the convergence to an architecture $\alpha$ whose neighbors in $\mathcal{N}(\alpha)$ all have a low

loss, which implies that $\alpha$ is a flat minima. This makes $\mathrm{mean}$ a valid choice. For a similar reason, $\mathrm{median}$ and $\mathrm{max}$ are also valid choices.

Setting $g(\cdot)$ as $\mathrm{mean}$ or $\mathrm{max}$ also aligns well with previous work on flat minima. Chaudhari et al. (2017) propose an objective function for training neural networks so that flat minima are preferred during optimization. Their objective can be interpreted as a weighted average of the (transformed) function values of data points around the local minima, which inspires us to consider $\mathrm{mean}$ as one of the choices for $g(\cdot)$. Keskar et al. (2017) use the largest function value that can be attained in the neighborhood of a local minimum to characterize how sharp the minimum is, which leads us to set $g(\cdot)$ as $\mathrm{max}$.

- Variance.

  For an architecture $\alpha$, we can measure its flatness with the variance (standard deviation) of the performance of its neighbors in $\mathcal{N}(\alpha)$. Let $\sigma(f(\mathcal{N}(\alpha)))$ denote the standard deviation of the performance (*e.g.*, validation loss) of architectures in $\mathcal{N}(\alpha)$. But simply minimizing $\sigma(f(\mathcal{N}(\alpha)))$ can only result in an $\alpha$ with a flat neighborhood, but cannot guarantee that $\alpha$ is a local minimum (*e.g.*, have a low validation loss). So we propose the following variance-based aggregation function $g\left(f(\mathcal{N}(\alpha))\right) = f(\alpha) + \lambda\sigma(f(\mathcal{N}(\alpha)))$ that takes both the performance of $\alpha$ and the flatness of its neighborhood into account, where $\lambda$ is a hyper-parameter to balance the performance $f(\alpha)$ and the flatness $\sigma(f(\mathcal{N}(\alpha)))$.

## B.2 AGGREGATION FUNCTION IN DIFFERENTIABLE ARCHITECTURE SEARCH

When applying our formulation to differentiable NAS methods, $g(\cdot)$ needs to be differentiable, which immediately rules out $\mathrm{median}$. Our default choice is $\mathrm{mean}$ and we provide an outline of NA-DARTS using $\mathrm{mean}$ in the main text.

Both $\mathrm{mean}$ and the variance-based aggregation function are differentiable. We prefer $\mathrm{mean}$ because it requires fewer GPU memory. Theoretically, when computing $\nabla_\alpha g\left(f(\mathcal{N}(\alpha))\right)$, we need to keep all architectures in $\mathcal{N}(\alpha)$ in GPU. But when $g(\cdot) = \mathrm{mean}$, we can compute $\nabla_\alpha f(\alpha')$ separately for each neighbor $\alpha' \in \mathcal{N}(\alpha)$. Since PyTorch (Paszke et al., 2019) automatically accumulates the gradient in multiple backward passes, computing $\nabla_\alpha f(\alpha')$ separately is equivalent as computing $\nabla_\alpha \mathrm{mean}\left(f(\mathcal{N}(\alpha))\right)$. Therefore, when using $\mathrm{mean}$, we only need to keep one architecture in GPU. This requires much fewer GPU memory than the variance-based aggregation function.

We prefer $\mathrm{mean}$ over $\mathrm{max}$ due to its superior empirical performance. When using $\mathrm{max}$, Eq. 5 becomes a minimax optimization problem and one can approximate the gradient of the objective using Danskin's Theorem (Danskin, 1967). Same as $\mathrm{mean}$, $\mathrm{max}$ also only needs to keep one architecture in GPU (see following text for more details).

---

**Algorithm 2** Neighborhood-Aware DARTS

---

**Input**: Number of steps $T$. Number of neighbors $n_{\mathrm{nbr}}$. Initial architecture $\alpha$ and weights $w$.
**for** $t = 1, 2, \ldots, T$ **do**
    Sample a batch of training data $X_{\mathrm{train}}$ and a batch of validation data $X_{\mathrm{val}}$.
    Sample $n_{\mathrm{nbr}}$ neighboring architectures of $\alpha$: $\mathcal{N}(\alpha)$.
    **if** $g(\cdot) == \mathrm{max}$ **then**
        Compute $\bar{\alpha} = \arg\max_{\alpha'\in\mathcal{N}(\alpha)} \mathcal{L}_{\mathrm{val}}(w, \alpha')$ on $X_{\mathrm{val}}$.
        Compute $\nabla_\alpha \mathcal{L}_{\mathrm{val}}(w, \bar{\alpha})$ on $X_{\mathrm{val}}$; update $\alpha$ by descending $\nabla_\alpha \mathcal{L}_{\mathrm{val}}(w, \bar{\alpha})$.
    **else if** $g(\cdot) == \mathrm{mean}$ **then**
        Compute $\nabla_\alpha \frac{\sum_{\alpha'\in\mathcal{N}(\alpha)} \mathcal{L}_{\mathrm{val}}(w,\alpha')}{|\mathcal{N}(\alpha)|}$ on $X_{\mathrm{val}}$; update $\alpha$ by descending $\nabla_\alpha \frac{\sum_{\alpha'\in\mathcal{N}(\alpha)} \mathcal{L}_{\mathrm{val}}(w,\alpha')}{|\mathcal{N}(\alpha)|}$.
    **end if**
    Compute $\nabla_w \mathcal{L}_{\mathrm{train}}(w, \alpha)$ on $X_{\mathrm{train}}$; update $w$ by descending $\nabla_w \mathcal{L}_{\mathrm{train}}(w, \alpha)$.
**end for**
Derive the final architecture based on the learned $\alpha$.

---

### B.2.1 USING $\mathrm{max}$ IN NA-DARTS

For completeness, we describe details of using $\mathrm{max}$ in NA-DARTS. After setting $g(\cdot)$ as $\mathrm{max}$, Eq. 5 becomes a minimax optimization. According to Danskin's Theorem (Danskin, 1967), we

---

**Algorithm 3** Neighborhood-Aware Random Search

---

**Input**: Number of steps $T$. Number of neighbors $n_{\text{nbr}}$.
**for** $t = 1, 2, \ldots, T$ **do**
    Randomly sample an architecture from $\mathcal{A}$: $\alpha$.
    Sample $n_{\text{nbr}}$ neighboring architectures of $\alpha$: $\mathcal{N}(\alpha)$.
    Train the $n_{\text{nbr}}$ architectures and compute $g\left(f(\mathcal{N}(\alpha))\right)$.
    Let $\alpha^* = \alpha$ if $g\left(f(\mathcal{N}(\alpha))\right) < g\left(f(\mathcal{N}(\alpha^*))\right)$.
**end for**
Return the optimal architecture $\alpha^*$.

---

can approximate the gradient $\nabla_\alpha \max_{\alpha' \in \mathcal{N}(\alpha)} \mathcal{L}_{\text{val}}(w^*(\alpha'), \alpha')$ with $\nabla_\alpha \mathcal{L}_{\text{val}}(w^*(\bar{\alpha}), \bar{\alpha})$, where $\bar{\alpha}$ is the maximizer of the inner maximization problem $\max_{\alpha' \in \mathcal{N}(\alpha)} \mathcal{L}_{\text{val}}(w^*(\alpha'), \alpha')$. In practice, $w^*(\alpha')$ is approximated by the current network weights $w$. To compute the maximizer $\bar{\alpha}$, we simply compute the validation loss of each sampled neighboring architecture and choose the maximum one. We provide an outline of NA-DARTS in Algorithm 2, where we include steps for both cases ($g(\cdot) = \max$ or $g(\cdot) = \text{mean}$). As can be seen from Algorithm 2, when using $\max$, we only need to keep one architecture ($\bar{\alpha}$) in GPU during the gradient computation.

Solving the inner maximization problem $\max_{\alpha' \in \mathcal{N}(\alpha)} \mathcal{L}_{\text{val}}(w^*(\alpha'), \alpha')$ is the process of finding the worst-performing neighbor of $\alpha$ in its neighborhood. Sampling neighbors with the additive representation of neighbors (Eq. 6) might not always result in a neighbor $\alpha'$ that performs worse than $\alpha$. So, we develop the following *multiplicative representation* of neighboring architectures. The multiplicative representation allows us to sample $\alpha'$ by changing a subset of operations in $\alpha$ to the zero operation or skip connection such that $\alpha'$ has a higher probability to perform worse than $\alpha$. Let edge $(i, j)$ be an edge to be perturbed and $r^{(i,j)}$ be a $m$-dim one-hot vector with $r_l^{(i,j)} = 1$ and $r_k^{(i,j)} = 0 (1 \leq k \leq m, k \neq l)$. We restrict $l$ to be either the index of the zero operation or skip connection. With the one-hot vector $r^{(i,j)}$, $\alpha'^{(i,j)}$ is computed as:

$$\alpha_k'^{(i,j)} = \frac{r_k^{(i,j)} \alpha_k^{(i,j)}}{\sum r_k^{(i,j)} \alpha_k^{(i,j)}}. \tag{7}$$

Under the multiplicative representation, $\alpha'^{(i,j)}$ has the same value as $r^{(i,j)}$, which indicates that the edge $(i, j)$ after perturbation chooses either the zero operation or skip connection. We empirically observe that $\max$ works better with the multiplicative representation than additive representation.

## C   ASSUMPTION JUSTIFICATION

### C.1   EXPERIMENTAL SETUP

We describe the detailed setup of our assumption justification experiments in Sec. 3.3. NAS-Bench-201 (Dong & Yang, 2020) provides a simulated environment for NAS experiments by conducting a thorough evaluation of all the candidate architectures (cells) in a pre-defined cell search space on three datasets: CIFAR-10 (Krizhevsky et al., 2009), CIFAR-100 (Krizhevsky et al., 2009), and ImageNet-16-120 (Dong & Yang, 2020). It contains the validation error (accuracy) of all the candidate architectures on CIFAR-10 after every training epoch, and the final test error on CIFAR-10, CIFAR-100, and ImageNet-16-120. ImageNet-16-120 is a subset and downsampled version of ImageNet (Russakovsky et al., 2015) and contains about 158K images divided into 120 classes.

In our experiments, we set the distance threshold $d$ to 1, so each architecture in the NAS-Bench-201 search space has 25 neighbors including itself. We search on CIFAR-10 and evaluate the found architectures on all three datasets, *i.e.*, $f(\alpha)$ is defined as the validation error on CIFAR-10. It is common in NAS to use early stopping or budgeted training during search (Elsken et al., 2019; Li et al., 2020). So, we use the CIFAR-10-Validation error after the $90^{th}$ epoch in the experiments, unless otherwise stated. Results for other epochs (*e.g.*, $30^{th}$, $60^{th}$, $120^{th}$) lead to the same conclusion.

## C.2 FLAT MINIMA GENERALIZE BETTER

We provide results for other epochs to show that flat minima in the architecture space generalize better than sharp minima. Specifically, we conduct the same experiments as Sec. 3.3.1 (Table 1a in the main text) with the CIFAR-10-Validation error after the $30^{th}$, $60^{th}$ or $120^{th}$ epoch. As shown in Table 7, results for all epochs ($30^{th}$, $60^{th}$, $90^{th}$, $120^{th}$) demonstrate the same pattern: the average validation error on CIFAR-10 of flat minima and sharp minima are similar; however, the average test error of flat minima is consistently lower than sharp minima on all three datasets, especially on CIFAR-100 and ImageNet-16-120.

## C.3 AGGREGATED PERFORMANCE GIVES A BETTER RANKING OF ARCHITECTURES

We show that our criterion $g\left(f(\mathcal{N}(\alpha))\right)$ ranks architectures more accurately than the standard criterion $f(\alpha)$. To do that, we randomly sample 100 architectures from NAS-Bench-201 and rank these architectures according to our criterion $g\left(f(\mathcal{N}(\alpha))\right)$ or the standard criterion $f(\alpha)$, where $f(\cdot)$ is the validation error on CIFAR-10. Following Yu et al. (2020), we evaluate the estimated ranking with the Kendall's Tau metric (the higher the better), which measures the correlation between the estimated ranking and ground truth ranking of architectures. The ground truth is obtained by sorting these architectures based on their test error. As the ground truth is specific to each dataset, we evaluate the estimated ranking on the three datasets separately.

We repeat the experiments for 10 times and report the mean and standard deviation of the Kendall's Tau value. Table 1b (main text) shows the ranking estimation results when $g(\cdot) = \mathrm{mean}$. We provide the results for all the aggregation functions in Table 8. For the variance-based aggregation function, we set $\lambda$ to 1.0. All aggregation functions except $\mathrm{max}$ result in an more accurate ranking estimation of architectures than the standard criterion $f(\alpha)$.

## C.4 AGGREGATED PERFORMANCE FINDS FLAT MINIMA

We conduct quantitative analysis to show that optimizing the proposed criterion, *i..e*, the aggregated performance over the neighborhood, finds flat minima. We select 100 architectures from NAS-Bench-201 with the lowest validation error (baseline criterion) on CIFAR-10, and another 100 architectures with the lowest aggregated validation error (proposed criterion).

We measure the flatness of an architecture using neighborhood variance, where smaller variance indicates flatter neighborhood. We summarize the neighborhood variance and test error of the found architectures in Table 9. We observe that optimizing the mean validation error ('Ours - mean') can successfully help us find flat minima, as the found architectures have a much smaller neighborhood variance than those found by the baseline criterion, and also achieve lower classification error on all three datasets.

We also notice that when $g(\cdot) = \mathrm{max}$, the found architectures are not flat minima. Although these architectures have a flat neighborhood (low neighborhood variance), their classification performance is worse than architectures found by the baseline criterion. We think this is because when using $\mathrm{max}$, the objective $g\left(f(\mathcal{N}(\alpha))\right)$ only considers the flatness of the neighborhood, but fails to characterize how well the architecture $\alpha$ performs.

# D NA-RS

**Experimental setup.** An outline of NA-RS is provided in Algorithm 3. Same as the setup in the assumption justification experiments, we search on CIFAR-10 and evaluate on CIFAR-10 (Krizhevsky et al., 2009), CIFAR-100 (Krizhevsky et al., 2009), and ImageNet-16-120 (Dong & Yang, 2020). The number of search steps $T$ in NA-RS is set to 100. For fair comparison, the standard random search (baseline; denoted as 'RS') is run for $T \cdot n_{\mathrm{nbr}}$ steps, so that RS and NA-RS train and evaluate the same number of architectures. We set the distance threshold $d$ to 1, so the neighborhood contains 25 architectures including the reference architecture itself. We set $n_{\mathrm{nbr}}$ to 10 unless otherwise stated.

**Ablation study.** We provide an ablation study of the aggregation function in NA-RS in Table 10 and an ablation study of $n_{\mathrm{nbr}}$ in Table 11. We see from Table 10 that $\mathrm{mean}$ and $\mathrm{median}$ achieve the best performance among all the choices for $g(\cdot)$. $\mathrm{max}$ performs the worst, which is consistent

Table 7: Average error of flat-minima architectures and sharp-minima architectures. "CIFAR-10-Validation" refers to the average validation error on CIFAR-10 used in search. CIFAR-10, CIFAR-100 and ImageNet-16-120 refer to the average test error on each dataset. Flat minima and sharp minima obtain a similar validation error on CIFAR-10. However, flat minima consistently achieves lower test error than sharp minima on all three datasets.

(a) $f(\alpha)$ = CIFAR-10-Validation error after the $30^{th}$ epoch.

|  | CIFAR-10-Validation | CIFAR-10 | CIFAR-100 | ImageNet-16-120 |
|---|---|---|---|---|
| Flat minima | 18.39 | **6.33** | **29.15** | **55.52** |
| Sharp minima | 18.45 | 6.67 | 30.10 | 56.18 |

(b) $f(\alpha)$ = CIFAR-10-Validation error after the $60^{th}$ epoch.

|  | CIFAR-10-Validation | CIFAR-10 | CIFAR-100 | ImageNet-16-120 |
|---|---|---|---|---|
| Flat minima | 16.15 | **6.28** | **29.15** | **55.51** |
| Sharp minima | 16.43 | 6.91 | 30.56 | 57.31 |

(c) $f(\alpha)$ = CIFAR-10-Validation error after the $90^{th}$ epoch.

|  | CIFAR-10-Validation | CIFAR-10 | CIFAR-100 | ImageNet-16-120 |
|---|---|---|---|---|
| Flat minima | 14.55 | **6.23** | **28.90** | **55.17** |
| Sharp minima | 14.57 | 6.66 | 30.00 | 56.41 |

(d) $f(\alpha)$ = CIFAR-10-Validation error after the $120^{th}$ epoch.

|  | CIFAR-10-Validation | CIFAR-10 | CIFAR-100 | ImageNet-16-120 |
|---|---|---|---|---|
| Flat minima | 12.67 | **6.13** | **28.59** | **55.11** |
| Sharp minima | 12.81 | 6.33 | 29.28 | 55.53 |

Table 8: Kendall's Tau (rank correlation) obtained by the standard criterion $f(\alpha)$ (baseline) and our criterion $g\left(f(\mathcal{N}(\alpha))\right)$ with different choices of $g(\cdot)$.

|  | CIFAR-10 | CIFAR-100 | ImageNet-16-120 |
|---|---|---|---|
| Baseline | $0.66 \pm 0.03$ | $0.66 \pm 0.02$ | $0.64 \pm 0.03$ |
| Ours - mean | $\mathbf{0.76 \pm 0.03}$ | $\mathbf{0.77 \pm 0.03}$ | $\mathbf{0.74 \pm 0.03}$ |
| Ours - median | $0.72 \pm 0.03$ | $0.72 \pm 0.03$ | $0.69 \pm 0.03$ |
| Ours - max | $0.53 \pm 0.05$ | $0.54 \pm 0.05$ | $0.56 \pm 0.05$ |
| Ours - Variance | $0.72 \pm 0.02$ | $0.73 \pm 0.03$ | $0.71 \pm 0.02$ |

Table 9: Neighborhood variance and test error of architectures found by by the standard criterion $f(\alpha)$ (baseline) and our criterion $g\left(f(\mathcal{N}(\alpha))\right)$ with different choices of $g(\cdot)$. Architectures found by the mean validation error ('Ours - mean') have a much smaller neighborhood variance than those found by the baseline criterion, and also achieve lower classification error on all three datasets.

|  | Neighbor-Var | CIFAR-10 | CIFAR-100 | ImageNet-16-120 |
|---|---|---|---|---|
| Baseline | 5.58 | 6.45 | 29.45 | 55.79 |
| Ours - mean | 2.71 | **6.09** | **28.32** | **54.75** |
| Ours - median | 4.05 | 6.21 | 28.74 | 55.08 |
| Ours - max | 1.83 | 6.66 | 29.82 | 56.31 |
| Ours - Variance | 2.47 | 6.35 | 29.06 | 55.52 |

with the conclusion in Table 8. As shown in Table 11, performance obtained by $n_{nbr} = 10$ is close to $n_{nbr} = 25$, which indicates that sampling a subset of neighbors is a good approximation for the entire neighborhood.

Table 10: Ablation study on the aggregation function in NA-RS. mean and median yield the lower test error among all the choices for $g(\cdot)$.

|  | CIFAR-10 | CIFAR-100 | ImageNet-16-120 |
|---|---|---|---|
| NA-RS - mean | $6.39 \pm 0.71$ | $28.68 \pm 1.75$ | $55.02 \pm 1.71$ |
| NA-RS - median | $6.20 \pm 0.35$ | $28.33 \pm 1.22$ | $54.72 \pm 0.96$ |
| NA-RS - max | $6.73 \pm 0.71$ | $29.70 \pm 1.61$ | $56.96 \pm 2.09$ |
| NA-RS - Variance | $6.65 \pm 0.97$ | $29.06 \pm 1.97$ | $55.48 \pm 2.41$ |

Table 11: Ablation study on $n_{\text{nbr}}$ in NA-RS. Sampling a subset of neighbors ($n_{\text{nbr}} = 10$) is a good approximation for the entire neighborhood ($n_{\text{nbr}} = 25$).

|  |  | CIFAR-10 | CIFAR-100 | ImageNet-16-120 |
|---|---|---|---|---|
| NA-RS - mean | $n_{\text{nbr}} = 10$ | $6.39 \pm 0.71$ | $28.68 \pm 1.75$ | $55.02 \pm 1.71$ |
|  | $n_{\text{nbr}} = 25$ | $6.24 \pm 0.39$ | $28.24 \pm 1.25$ | $54.74 \pm 1.73$ |
| NA-RS - median | $n_{\text{nbr}} = 10$ | $6.20 \pm 0.35$ | $28.33 \pm 1.22$ | $54.72 \pm 0.96$ |
|  | $n_{\text{nbr}} = 25$ | $6.18 \pm 0.38$ | $28.20 \pm 1.27$ | $54.40 \pm 0.98$ |

## E NA-DARTS

### E.1 EXPERIMENTAL SETUP

Following DARTS (Liu et al., 2019), we search on CIFAR-10 (Krizhevsky et al., 2009) and evaluate on CIFAR-10 (Krizhevsky et al., 2009), CIFAR-100 (Krizhevsky et al., 2009) and ImageNet (Russakovsky et al., 2015). We use exactly the same setup as DARTS Liu et al. (2019), including the cell search space, hyper-parameters, such as the learning rate and weight decay factor, and other experimental details. We split the training images in CIFAR-10 into two subsets of equal size, which are used as the training and validation images during search. We construct a network of 8 cells with an initial channel number as 16 and train the network for 50 epochs to learn $\alpha$.

After the search is done, we derive the final architecture from the learned $\alpha$ using exactly the same procedure as DARTS. When evaluating the found architecture on CIFAR-10 and CIFAR-100, we build a network of 20 cells and train it for 600 epochs with batch size 96 and cutout (DeVries & Taylor, 2017). For our NA-DARTS, We set the initial number of channels of the network such that it has a similar network size with DRATS and contains around 3M parameters.

When evaluating on ImageNet, we build a network of 14 cells. Following DARTS, the network is trained for 250 epochs with batch size 128. We set the initial number of channels such that the number of multiply-add operations in the network is fewer than 600M when the input is $224 \times 224$. Some NAS methods use a different training setup to train the found architecture on ImageNet. For example, DARTS+ (Liang et al., 2019) trains for 800 epochs and P-DARTS (Chen et al., 2019) uses a large batch size 1024 (need 8 V100 GPUs, infeasible to us). For fair comparison, we retrain the found architecture reported by the authors in their paper using the same training setup as DARTS.

For our NA-DARTS, we sample a subset of 10 neighbors in each step, *i.e.*, $n_{\text{nbr}} = 10$. The distance threshold $d$ for neighborhood can be interpreted as the number of edges to be perturbed. As each cell in the DARTS search space has 14 edges, we set $d$ to 6. The noise threshold $\epsilon$ in the additive representation is set to 0.1. All experiments are performed on a NVIDIA GeForce RTX 2080 Ti GPU.

### E.2 ABLATION STUDY

**Aggregation function.** We report the performance of NA-DARTS when using mean or max as the aggregation function in Table 12a. We observe that mean outperforms max, which is consistent with the conclusion in Table 8. We also notice that mean consumes a longer search time than max. This is because when using mean, we need to back-propagate through every sampled neighboring

architecture $\alpha'$, while we only need to back-propagate through one neighboring architecture $\bar{\alpha}$ when using max.

**Distance threshold.** We study the impact of the distance threshold of $d$ in Table 12b, where we observe $d = 6$ achieves the best performance and $d = 4$ performs similarly with $d = 6$. Recall that the distance threshold $d$ can be interpreted as the number of edges to be perturbed and the cell in the DARTS search space has 14 edges. We empirically find that when $d$ becomes larger that 6, the neighborhood becomes too large and the performance drops.

Table 12: Ablation study of NA-DARTS.

(a) Impact of aggregation function.

|  | Test Error (%) | | Param (M) | Search Cost (GPU days) |
|---|---|---|---|---|
|  | CIFAR-10 | CIFAR-100 | | |
| max | $2.80 \pm 0.10$ | $16.89 \pm 0.31$ | 3.1 | 0.5 |
| mean | $2.63 \pm 0.12$ | $16.48 \pm 0.13$ | 3.2 | 1.1 |

(b) Impact of $d$.

|  | Test Error (%) | | Param (M) |
|---|---|---|---|
|  | CIFAR-10 | CIFAR-100 | |
| $d = 2$ | $2.62 \pm 0.08$ | $16.90 \pm 0.45$ | 3.2 |
| $d = 4$ | $2.65 \pm 0.19$ | $16.56 \pm 0.36$ | 3.1 |
| $d = 6$ | $2.63 \pm 0.12$ | $16.48 \pm 0.13$ | 3.2 |

## F  LOSS LANDSCAPE VISUALIZATION

To qualitatively examine whether our NA-DARTS has found a flat minima, we plot the loss landscape of DARTS and NA-DARTS with the visualization strategy from Li et al. (2018). Let $\alpha$ denote the architecture found by DARTS or NA-DARTS. We compute the Hessian of the validation loss with respect to $\alpha$, and $v_0$ and $v_1$, which are the eigenvectors corresponding to the two largest eigenvalues of the Hessian matrix. Then we visualize the validation loss of the neighbors of $\alpha$ over the plane spanned by $v_0$ and $v_1$. Specifically, we compute the validation loss of the architecture $\alpha + \lambda_0 v_0 + \lambda_1 v_0$, where $\lambda_0$ and $\lambda_1$ are uniformly sampled from $[-1.0, 1.0]$. The loss values are visualized by the contour plots in Figure 2. We observe that the curvature of NA-DARTS at $(0, 0)$ (the found architecture $\alpha$) is much flaterr than that of DARTS.

We provide details of the neighboring architecture $\alpha' = \alpha + \lambda_0 v_0 + \lambda_1 v_0$, where we overload the plus sign (+) with the additive representation. Recall that $\alpha$ contains a set of variables representing the operation choice for each edge $(i, j)$: $\alpha = \{\alpha^{(i,j)}\}$. The eigenvectors $v_0$ and $v_1$ have the same dimension as $\alpha$ and then can be represented as $v_0 = \{v_0^{(i,j)}\}$ and $v_1 = \{v_1^{(i,j)}\}$. Let $q^{(i,j)} = \lambda_0 v_0^{(i,j)} + \lambda_1 v_1^{(i,j)}$. $\alpha'^{(i,j)}$ is then computed using the additive representation in Eq. 6 ($\alpha'^{(i,j)}_k = \frac{\alpha_k^{(i,j)} + q_k^{(i,j)}}{\sum_{k=1}^{n}(\alpha_k^{(i,j)} + q_k^{(i,j)})}$). The eigenvectors $v_0$ and $v_1$ are normalized so that the scale of the noise vector $q^{(i,j)}$ is controlled only by $\lambda_0$ and $\lambda_1$. We use the weights of $\alpha$ obtained in the search as an approximation for the weights of the neighbors $\alpha'$.

## G  CELL VISUALIZATION

We visualize the normal cell and reduction cell found by DARTS and our NA-DARTS in Figure 3. We observe that the normal cell found by our method NA-DARTS tend to be deeper than that found found by DARTS. Normal cells found by our NA-DARTS from different runs have a depth of 3 at most of the time, while normal cells found by DARTS mostly have a depth of 1 or 2. We also observe that the normal cell found by NA-DARTS contains more $5 \times 5$ convolution operations. Both of the reduction cells found by DARTS and NA-DARTS contain very few convolution operations. Most operations in the reduction cell do not have parameters, *e.g.*, pooling and skip-connection.

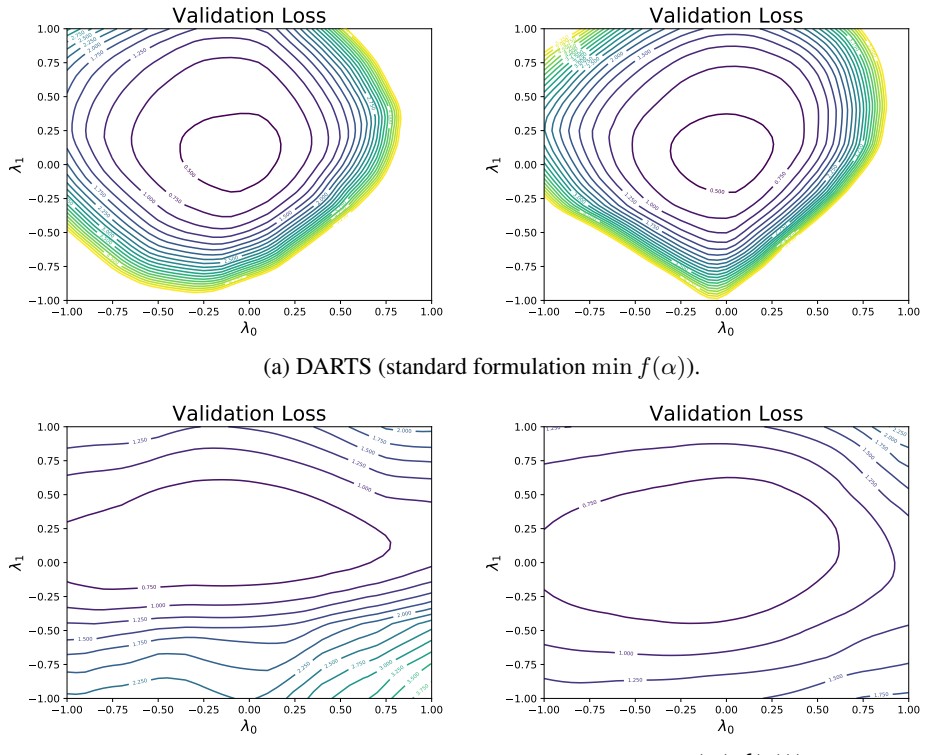

(a) DARTS (standard formulation $\min f(\alpha)$).

(b) NA-DARTS (neighborhood-aware formulation $\min g\left(f(\mathcal{N}(\alpha))\right)$).

Figure 2: Loss landscape visualization of the found architecture. The two plots in Figure 2a (Figure 2b) are generated from two independent runs of DARTS (NA-DARTS). The left plot in Figure 2a and Figure 2b are the same as the plots in Figure 1 in the main text. For the architecture found by DARTS (Figure 2a), we observe that the loss of its neighbors increase drastically as the magnitude of $\lambda_0$ or $\lambda_1$ increases. However, for the architecture found by our NA-DARTS (Figure 2b), the loss of its neighbors increases much slower. This shows that the architecture found by our NA-DARTS is a much flatter minimum than that found by DARTS.

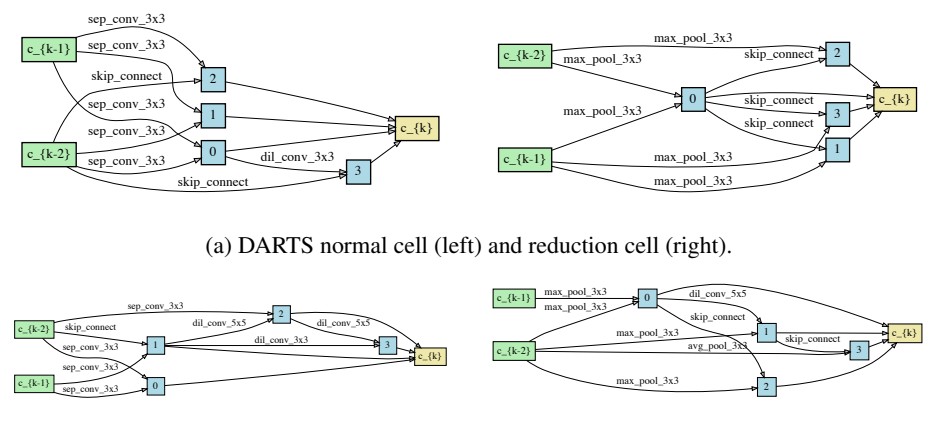

(a) DARTS normal cell (left) and reduction cell (right).

(b) NA-DARTS normal cell (left) and reduction cell (right).

Figure 3: Cell Visualization.

