# OpenReview forum: "Neighborhood-Aware Neural Architecture Search"
_ICLR.cc/2021/Conference — Reject_

### Official Review · AnonReviewer1 · 2020-10-28
**The results are good but novelty is not enough**

**Rating:** 4
**Confidence:** 5

**Review:**

Inspired by the concept that ``flat minima generalize better than sharp minima", this paper proposes to search the flat-minima architecture by considering the performance over the neighborhood architectures. A random search and differentiable search method based on the neighborhood principle are proposed, which achieve comparable results with SOTA NAS methods.

Concrete comments
1. The biggest concern lies in the main idea of the neighborhood-aware search formulation. It is known in many previous methods that the flat minima of the loss function of neural network training generalize better. This conclusion is also extended to the eigenspectrum of the Hessian of the validation loss with respect to the architectural parameters in [1]. However, this paper hypothesizes that similar architectures with similar performance point to a flat solution. How it can be proven that nearby architectures are located in neighborhood in the loss space should be clarified. This is of critical importance to support the main concept of the paper.
2. The proposed method is not so reasonable. The distance between architectures is defined as Eq. (2). However, many cases may fail with Eq. (2). For example, a 3x3 separable convolution is nearer to a 5x5 separable convolution, but further from the identity operation, which cannot be handled with Eq. (2). Moreover, the final operation is determined with the highest probability. [0.05, 0.6, 0.35] and [0.35, 0.6, 0.05] represent the same architecture, but different in Eq. (2).
3. Experimental results are not so convincing.
  - In Tab. 1 and 2, only test errors are reported but not Params or FLOPs. It is not clear whether advantages of the method actually exist. Normally, models should be compared with similar sizes.
  - The proposed method does not show evident advantages over others, e.g., worse or similar compared with PC-DARTS but taking much more cost, 10x than PC-DARTS.
  - The experiments are based on the DARTS-like search space. However this kind of search spaces hold high latency due to the complicated topology. What about the widely used MobileNet block-based search space?
4. The novelty is somewhat limited. As the flat solution has been explored in [1], the main contribution seems to be the newly defined loss function ensemble with similar architectures.

[1] Arber Zela, Thomas Elsken, Tonmoy Saikia, Yassine Marrakchi, Thomas Brox, and Frank Hutter. Understanding and robustifying differentiable architecture search. In ICLR, 2020.

---

> ### Author Response · Authors · 2020-11-25
> **Response to R1 - Part 1 (Novelty)**
>
> We thank R1 for the valuable feedback.
>
> **Novelty**
>
> While the flat minima idea has been explored in Zela et al., 2020, our method differs from theirs in the following aspects:
>
> (1) Our formulation encourages the search algorithm to **actively** looking for flat minima architectures from the search space, while their method does not change the normal optimization process but stops optimization before the curvature of the minimum becomes too sharp.
>
> (2) We use the aggregated performance $g \left(f(\mathcal{N}(\alpha)) \right)$ to identify flat minima, while they use the largest eigenvalue of the Hessian matrix.
>
> (3) Our formulation is applicable to sampling-based NAS methods, while their method is not immediately applicable since they require computing the Hessian matrix, which is difficult for sampling-based NAS.
>
> (4) We provide extensive analysis to verify that the flat minima assumption also holds true in the architecture space. This is novel and one of our contributions.
>
> (5) As shown in Table 6, our method (NA-DARTS) outperforms their method (DARTS-ES). Also, we conduct further experiments as suggested by R2, where we show that our neighborhood-aware formulation is orthogonal to their early stopping strategy. Combining the two methods yields further improvement (see NA-DARTS-ES in response to R2).

---

> > ### Author Response · Authors · 2020-11-25
> > **Response to R1 - Part 2**
> >
> > **Concerns about the neighborhood-aware formulation and flat minima assumption**
> >
> > There might be a misinterpretation in R1’s statement that “this paper hypothesizes that similar architectures with similar performance point to a flat solution”. This is not our assumption but simply describes the property of flatness, where similar inputs yield similar outputs. In our case, the input is the architecture and the output is the validation accuracy/loss.
> >
> > To measure the flatness of an architecture, we adopt the aggregated validation accuracy/loss over the neighborhood as our objective function (defined in Eq. 1; see Sec. 3.2 for more details). We validate that our objective function can indeed find flat minima in Appendix C.4.
> >
> > Furthermore, we verified the flat minima assumption through extensive experiments that a flat-minima architecture generalizes better than a sharp-minima architecture. For more details, please refer to Sec. 3.3 and Appendix C. Note these analyses are also acknowledged by R2 (“the authors verify it in Appendix C”) and R4 (“The paper supports their assumption with various empirical experiments and ablation studies”).
> >
> >
> > **Questions about Eq. 2**
> >
> > We thank R1 for pointing out the intuition that a 3x3 conv is closer to a 5x5 conv than the identity operation. We agree that taking this into account will further strengthen our method. This requires us to quantitatively compute the similarity/distance between any two operations, which is an open problem and will be an interesting future direction to extend our idea.
> >
> > We agree that a change in probability distributions does not always result in a different discrete architecture. This is common in differentiable NAS methods. For example, in DARTS, the architecture before and after a gradient update are different in the continuous probability space, but might still represent the same discrete architecture. Also, even if the two probability distributions represent the same discrete architecture, they are still informative as the probability change denotes the relative importance of operations.
> >
> >
> > **Params and FLOPS of architectures in Table 1&2**
> >
> > The results in Table 1&2 are obtained by experiments on NAS-Bench-201 (Dong & Yang, 2020).  It seems to be a convention for NAS-Bench-201 that researchers do not explicitly report the params/FLOPS of architectures when comparing search algorithms (e.g., Table 5 in the original NAS-Bench-201 paper). All the architectures have the same number of initial channels and cells, so most architectures should have similar params and FLOPS. In all the experiments on NAS-Bench-201, we directly query the benchmark for accuracy of each architecture without changing its size and training it.
> >
> > **Incremental Improvement**
> >
> > We agree that the improvement is not amazing. But our improvement is also not marginal given the fact that the cell search space used in DARTS has a narrow performance range as verified in Yang et al., 2020, e.g., the test error of most methods are within [25%, 27%] on ImageNet.
> > Taking two recent methods PC-DARTS [A] (ICLR 2020) and PVLL-NAS [B] (ICML 2020) for example, their improvement over DARTS on ImageNet are 1.0% and 1.1% respectively. Our improvement over DARTS on ImageNet is higher than both of them. Also, as shown in Table 6, our method can be combined with PC-DARTS to further improve the performance.
> >
> > [A] Yuhui Xu, Lingxi Xie, Xiaopeng Zhang, Xin Chen, Guo-Jun Qi, Qi Tian, Hongkai Xiong. PC-DARTS: Partial Channel Connections for Memory-Efficient Architecture Search. ICLR 2020.
> >
> > [B] Yanxi Li, Minjing Dong, Yunhe Wang, Chang Xu, Neural Architecture Search in A Proxy Validation Loss Landscape. ICML 2020.
> >
> >
> >
> > **DARTS search space**
> >
> > All the baselines we compare to in Table 4&5 only report results on the DARTS-like search space. To ensure consistent evaluation and fair comparison to previous work, we also evaluate our method on the DARTS search space. We agree that having results on the MobileNet block-based search space will be very useful, but due to limited resources, we leave that for the future work.

---

### Official Review · AnonReviewer2 · 2020-10-28
**An interesting idea for NAS, where the neighborhood of the model is considered.**

**Rating:** 6
**Confidence:** 4

**Review:**

** Summary
The authors proposed neighborhood-aware neural architecture search, where during the evaluation phase during search, the neighborhood of an architecture is considered. Specifically, when an architecture $\alpha$ is picked, its neighbors $\mathcal{N}(\alpha)$ all contribute to the performance validation. This is built upon the assumption that `` flat minima generalize better than sharp minima’’ and the authors verify it in Appendix C.
The authors conducted experiments on CIFAR-10/100 and ImageNet, and obtained promising improvements over the standard baselines.

** Clarify
1.	Towards "Due to the property of the total variation distance, when $d$ is an integer, the neighborhood contains all the cells that have at most $d$ edges associated with different operations from $\alpha$". As you reported, $\alpha$ is a collection of $\alpha^{(i,j)}$, where each $\alpha^{(i,j)}$ is a one hot vector “ But in DARTS, $\alpha^{i,j}$ is a distribution of all candidate operations. In this case, how to select the $\mathcal{N}(\alpha)$?

** Significance
Overall, I think the results are solid.
1.	There is a possible “ensemble’” baseline for your method. Let us take DARTS as an example. First, we independently train $n_{nbr}$ one-shot models. Each one-shot model has an individual $\alpha$. Then, we sample an architecture from the average of all $\alpha$’s. This could be another way to leverage neighborhood information and should be compared.
2.	Improvement not significant: In Table 4, compared to DARTS+, the improvement is 0.1/0.4 on CIFAR10/100. Compared to PDARTS, on the three datasets, the results between your method and PDARTS are almost the same. Therefore, you should apply your method to more recent advantaged methods to show that it is orthogonal or others.
3.    As you pointed in Section 2, (Zela et al 2020) also observed a strong correlation between the generalization error of the architecture found by DARTS. It is good to see the exploration in Table 6. I think the "NA-DARTS-ES" should be implemented to see whether your method and  (Zela et al 2020) are complementary to each other.
4.	The code should be released to reproduce your work.

== Post Rebuttal ==

I am satisfied with the response "ensemble baseline" and "NA-DARTS-ES". But my concern about "Improvement not significant" is not addressed, which is also mentioned by R1 and R4. I will remain my score as 6.

---

> ### Author Response · Authors · 2020-11-25
> **Response to R2**
>
> We thank R2 for the valuable feedback.
>
> As suggested by R2, we conduct the following two experiments:
>
> **Ensemble baseline**
>
> We average the $\alpha$ from 5 independent runs of DARTS and derive the final architecture. This architecture obtains 2.90% test error on CIFAR-10 and 18.07% test error on CIFAR-100. Our proposed NA-DARTS (2.63% on CIFAR-10 and 16.48% on CIFAR-100) easily outperforms this ensemble baseline on both datasets.
>
> In our NA-DARTS experiments, $n_\text{nbr}$=10. The reason why we only average 5 runs of DARTS are: (1) we did not have enough time to run DARTS for 10 times; and (2) the time to run DARTS for 5 times is already longer than running NA-DARTS for once (0.3x5=1.5 v.s. 1.1 GPU days).
>
> **NA-DARTS-ES**
>
> We thank R2 for suggesting this experiment. We empirically confirm that combining our method with the early stopping strategy in Zela et al. (2020) yields further improvement.  When implementing NA-DARTS-ES, we follow Zela et al. (2020) and stop the optimization based on the dominant eigenvalue of the Hessian matrix. Here are the results:
>
> |             |         CIFAR-10       |         CIFAR-100       |
> |-------------|:----------------------:|:-----------------------:|
> | DARTS       |          $4.13\pm0.98$ | $22.49\pm2.62$          |
> | NA-DARTS    |          $2.97\pm0.18$ | $18.86\pm0.49$          |
> | NA-DARTS-ES | $\mathbf{2.49\pm0.02}$ | $\mathbf{17.03\pm0.41}$ |
>
>
>
> **Clarification: how to select $\mathcal{N}(\alpha)$ for DARTS?**
>
> You are right that $\alpha^{(i, j)}$ is a probability distribution, as the architecture representation in DARTS is relaxed to be continuous. In this case, we sample the neighboring architectures by perturbing the probability distribution $\alpha^{(i, j)}$ with a randomly sampled noise vector. Please see Sec. 4.2.2 and Eq. 6 for more details. This strategy allows us to sample a set of neighboring architectures of $\alpha$ and the sampled architectures are represented as a differentiable function of $\alpha$.

---

### Official Review · AnonReviewer4 · 2020-10-28
**Well written but incremental improvements.**

**Rating:** 5
**Confidence:** 3

**Review:**

This paper introduces a searching framework of neural architectures search by modifying the objective function to optimize the aggregated performance over the neighborhood of an architecture. From the observation that flat-minima architecture $\alpha$ generalizes better than sharp-minima architecture (Zela et al. (ICLR2020), the author proposes the objective function considering the neighborhood to enforce the flat minima. The author supports their method by providing ablation studies and architecture performances from CIFAR-10/100, ImageNet, and NAS-BENCH-201.

Strength
1. Their objective function can easily be extended to existing NAS methods including Random Search and Gradient-Based methods.
2. The authors support their methods with various datasets such as CIFAR-10/100 and ImageNet
3. The paper supports their assumption with various empirical experiments and ablation studies.

Weakness
1.  The experiment results are incremental improvements (or par) to the existing NAS algorithms. By listing the recently published NAS literature on CIFAR-10: P-DARTs: 2.50%, DATA: 2.59%, SGAS: 2.66%, PC-DARTs: 2.57%, RandomNAS-NSAS: 2.64%.
2. Few experiments on NAS Benchmark. The experiments don't include NAS-Bench-201 (a popular benchmark for NAS) and small experiments with NAS-Bench-1SHOT1 only comparing with DARTs, PC-DARTs. And PC-DARTs result is better than NA-PC-DARTs according to Table 6.


Question
1. How do you select the number of neighbors? In the DARTs setup, the normal/reduce cell would have $(2+3+4+5)*7*2=196$. The possible neighboring architectures from a given subnetwork even with hamming distance 1 may be much larger than 10. Can you visualize the loss landscape based on various $n_{nbs}$?
2. Have you tried to verify your algorithm for NA-DARTs on NAS-Bench-201 which is the popular benchmark in recent NAS literature?
3. The reproduced ImageNet test error results on PC-DARTs and P-DARTs are higher than the reported test results. The reported test results on PC-DARTs outperforms the NA-DARTs result. Is the resulting discrepancy related to hyperparameter tuning? Can you clarify this?

Overall, this paper proposes a neighborhood aware objective function that can be adopted in the existing NAS search. Despite the paper well written, the performance of their methodology is incremental improvements (or par) among various existing NAS algorithms. I would recommend a marginally below acceptance threshold for now but may change the decision based on the rebuttal.

Reference:
1. Xu, Y et al. (2019, September). PC-DARTS: Partial Channel Connections for Memory-Efficient Architecture Search. ICLR2020
2. Chen, X et al. Progressive differentiable architecture search: Bridging the depth gap between search and evaluation. ICCV2019
3. Chang, J et al. DATA: Differentiable ArchiTecture Approximation. Neurips2019
4. Li, G et al. SGAS: Sequential Greedy Architecture Search. CVPR2020
5. Zhang, M et al. Overcoming Multi-Model Forgetting in One-Shot NAS with Diversity Maximization. CVPR2020

---

> ### Author Response · Authors · 2020-11-25
> **Response to R4**
>
> We thank R4 for the valuable feedback.
>
> **Incremental Improvement**
>
> We agree that the improvement is not amazing. But our improvement is also not marginal given the fact that the cell search space used in DARTS has a narrow performance range as verified in Yang et al., 2020, e.g., the test error of most methods are within [25%, 27%] on ImageNet.
> Taking two recent methods PC-DARTS [A] (ICLR 2020) and PVLL-NAS [B] (ICML 2020) for example, their improvement over DARTS on ImageNet are 1.0% and 1.1% respectively. Our improvement over DARTS on ImageNet is higher than both of them. Also, as shown in Table 6, our method can be combined with PC-DARTS to further improve the performance.
>
> [A] Yuhui Xu, Lingxi Xie, Xiaopeng Zhang, Xin Chen, Guo-Jun Qi, Qi Tian, Hongkai Xiong. PC-DARTS: Partial Channel Connections for Memory-Efficient Architecture Search. ICLR 2020.
>
> [B] Yanxi Li, Minjing Dong, Yunhe Wang, Chang Xu, Neural Architecture Search in A Proxy Validation Loss Landscape. ICML 2020.
>
> **Experiments on NAS Benchmarks and comparison to other baselines**
>
> We have conducted extensive experiments on NAS-Bench-201, including evaluating our NA-RS algorithm (Table 2&10&11) and justifying the flat minima assumption (1&7&8&9). We also compare to many baselines other than DARTS/PC-DARTS, on both CIFAR-10/100 and ImageNet in Table 4&5.
>
> We agree that evaluating our method on NAS-Bench-1SHOT1/NAS-Bench-201 would be useful, but most baselines (e.g., PC-DARTS) in Table 4&5 only report results on the DARTS search space. The DARTS search space is larger than the one used in NAS-Bench-1SHOT1 and NAS-Bench-201. To ensure consistent evaluation and fair comparison to previous work, we believe evaluating on the DARTS search space is sufficient to validate our method.
>
> **PC-DARTS in Table 6**
>
> PC-DARTS is **not** better than NA-PC-DARTS according to Table 6. Our NA-PC-DARTS outperforms PC-DARTS by 0.72% on CIFAR-100 and performs similarly to PC-DARTS on CIFAR-10. This result is positive and shows that our idea can be combined with PC-DARTS and gives further improvement for generalizing to different datasets.
>
> **ImageNet results of PC-DARTS and P-DARTS**
>
> PC-DARTS (Xu et al., 2019)  and P-DARTS (Chen et al., 2019) use a large batch size 1024 (need 8 GPUs, infeasible cost for our research group), while DARTS uses a batch size of 128. They also use different hyperparameters from DARTS, e.g. a different initial learning rate and weight decay. For fair comparison, we retrain the found architecture reported by the authors in their paper using the same training setup as DARTS.
>
> **How do you select the number of neighbors?**
>
> Yes, in the DARTS search space, the neighborhood of an architecture is large. Using all the neighbors is computationally prohibitive. We set $n_\text{nbr}$ to 10 as it empirically works well and the search cost is reasonable.

---

### Official Review · AnonReviewer3 · 2020-10-29
**Official Blind Review #3**

**Rating:** 6
**Confidence:** 4

**Review:**

This paper proposes a neighbor-aware method in the neural architecture search (NAS). The paper states that by optimizing a neighbor of the neural network, it can search the result in a flat-minima, which is more stable than the sharp minima. The experiment results in further support that the proposed NA-RS and NA-DARTS outperform the current SOTA in various tasks.

Overall, I think the paper raises an interesting opinion, which stresses that during the NAS process, a stable flat-minima has a better generalization ability. The definition of the neighbor of the neural architecture is also interesting. Below are some detailed points of my opinion about this paper.

Pros:
1. The paper compares with a lot of NAS methods and outperforms most of them.
2. Stabilize the search result can reach a better performance seems reasonable for me.
3. The ablation study in the appendix further shows the performance under different aggregate functions and distances, which makes the experiment more convincing.

Cons:
1. From the paper's statement, sample-based NAS is easier to define the neighbor than the gradient-based one. However, from the empirical result, the sample-based NAS result is relatively weak compared with the differential based. DARTS also provides the result of the random sampling on their search space. It is interesting to see whether NA-RS can outperform the naive random sampling and even catch up with some other NAS method in DARTS's setting and search space. Currently, I think only test the sample-based method on NAS-Bench-201 is a little bit toy.

2. PC-DARTS has reported its ImageNet performance, top1 error, 24.2%. I am a little bit confused about why the author here uses their implement result with a lower performance. In addition,  it is interesting that the author states that they can combine PC-DARTS with their neighbor-aware. However, they only show an S3 search space result in the appendix, it would be more convincing to show the neighbor-aware method's generalization ability if you can apply it on the standard search space and directly compare with PC-DARTS's public performance. Otherwise, the 'SOTA' statement looks not very strong for me.

================================After Response==============================
1. I agree that different DARTS paper usually uses some different settings, the lower PC-DARTS performance in the paper could be reasonable.
2. Search Space design sometimes can greatly impact the result of the final result, but it makes sense to modify the search space for a better result. I am glad that you state the situation of the performance on the original DARTS space.
3. I also read other reviewers' opinions. Based on the author's response and my previous rating, I decided to keep this score. This is an acceptable paper, but still has something to do in the future, like a more comprehensive experiment part.

---

> ### Author Response · Authors · 2020-11-25
> **Response to R3**
>
> We thank R3 for the valuable feedback.
>
> **ImageNet performance**
>
> Some NAS methods use a different training setup to train the found architecture on ImageNet. For example, DARTS+ (Liang et al., 2019) trains for 800 epochs while DARTS only trains for 250 epochs. PC-DARTS (Xu et al., 2019)  and P-DARTS (Chen et al., 2019) use a large batch size 1024 (need 8 GPUs, infeasible cost for our research group), while DARTS uses a batch size of 128. For fair comparison, we retrain the found architecture reported by the authors in their paper using the same training setup as DARTS.
>
> **NA-PC-DARTS on the DARTS search space**
>
> We choose the S3 space as it is a more challenging search space for DARTS and its variants, since Zela et al. (2020) shows that on S3, DARTS yields degenerate architectures with very poor test performance. On S3, our idea can be combined with PC-DARTS and gives further improvement, i.e., NA-PC-DARTS outperforms both NA-DARTS and PC-DARTS in Table 6.
>
> As suggested by R3, we apply our neighborhood-aware formulation to PC-DARTS on the standard DARTS search space. We find that NA-PC-DARTS performs similarly to our NA-DARTS on the DARTS search space. Our experiments on S3 search space provide a more complete picture of how NA-PC-DARTS works.
>
> **Sample-based NAS on DARTS search space**
>
> We agree that evaluating the sample-based method on more search spaces would be helpful. Due to limited computational resources and time, we haven’t been able to obtain the results on the DARTS search space. We leave that for future work.

---

### Decision · Program_Chairs · 2021-01-07
**Final Decision**

**Decision:**

Reject

**Comment:**

This paper proposes a new NAS methods that when doing architecture search, returns flat minima using based on a notion of distance defined for two cells (Eq. (2)). Authors then evaluation the effectiveness of the proposed methods against prior work on several benchmarks.

As authors have discussed in the paper, the idea of using flatness notion in architecture search is not new and has been first proposed by Zela et al 2020. This paper is building on Zela et al 2020 but the proposed algorithm is novel and different than Zela et al 2020. Even though the introduced algorithm is interesting, there are several concerns/areas of improvements:

1- The proposed method's performance is highly dependent to the notion of distance defined in eq. (2). However, the current choice is not well-motived and does not seem like a well-thought-out choice. See for example the issue raised by R1. I think authors need to spend more time on this choice. One other option is to meta-learn the vector representation of each operation.

2- All reviewers agree that the improvements marginal and in some cases not statistically significant. Authors have responded by arguing that this is typical for this area of research. I don't find this answer satisfying. For example, consider P-DARTS (Chen et al., 2019). P-DARTS improves over NA-DARTS (the proposed method) on CIFAR-10 and ImageNet and on CIFAR-100 they are on par given the standard deviation of NA-DARTS (see Tables 4 and 5). Moreover, the search cost of P-DART is 0.27% of NA-DARTS (Table 4). So P-DARTS has clear advantage over NA-DARTS.

Given the above issues, I recommend rejecting the paper. I hope authors would take feedbacks from the reviewing process into account to improve the paper and resubmit.